# Salivary Cortisol in Guide Dogs

**DOI:** 10.3390/ani13121981

**Published:** 2023-06-14

**Authors:** Enrique De la Fuente-Moreno, Pedro Paredes-Ramos, Apolo Carrasco-García, Bertha Hernandez-Cruz, Mayvi Alvarado, Claudia Edwards

**Affiliations:** 1Doctorado en Neuroetología, Xalapa Universidad Veracruzana, Xalapa 91190, Mexico; efm121@yahoo.com.mx; 2Facultad de Medicina Veterinaria y Zootecnia, Universidad Veracruzana, Veracruz 91190, Mexico; acarrasco@uv.mx (A.C.-G.); behernandez@uv.mx (B.H.-C.); 3Instituto de Neuroetología, Xalapa Universidad Veracruzana, Xalapa 91190, Mexico; malvarado@uv.mx; 4Facultad de Medicina Veterinaria y Zootecnia, Universidad Nacional Autónoma de México, Mexico City 04510, Mexico; dra.edwards@yahoo.com

**Keywords:** cortisol, welfare, guide dogs, companion dogs

## Abstract

**Simple Summary:**

We compared cortisol levels in the saliva of guide dogs and dogs that were trained as such but became companion dogs during a period of social isolation and exposure to a gunshot sound. The results showed that cortisol levels were higher in guide dogs than in companion dogs throughout the test. No changes were observed as a consequence of social isolation or exposure to the gunshot. This suggests that guide dogs maintain higher levels of basal cortisol compared with companion dogs, which could be associated with cognitive processes derived from working as guide dogs.

**Abstract:**

Guide dogs work for extended periods and are exposed to multiple environmental stimuli that could lead to higher stress compared with companion dogs. Cortisol is the main hormone associated with stress in most mammals. This study included seven guide dogs and seven same-breed dogs that were trained as guide dogs but became companion dogs to compare their salivary cortisol levels before, during, and after a period of social isolation and exposure to a 110-decibel gunshot sound. Each dog was left alone in an empty room for 60 min. After 15 min, the dogs were exposed to the sound. We collected four saliva samples from each dog. The first one was taken 5 min before starting the social isolation period, and the following ones at 15, 30, and 45 min after the test started. A two-way ANOVA was used to compare the group effect and the time effect during isolation and noise exposure. The results showed higher levels of cortisol in the guide dogs compared with the companion dogs throughout the test. No differences were found in time or in the interaction between time and group. This suggests that being a guide dog increases levels of basal cortisol when compared with dogs that live as companion animals and family members.

## 1. Introduction

The use of dogs as guides for blind or visually impaired people began in Germany during World War I, when it is estimated that between 5 and 8% of the soldiers’ injuries were to the eyes. In 1916, Dr. Gerhard Stalling founded the first guide dog training school (GDTS) using German Shepherd dogs [1]. Currently, there are many GDTSs around the world, where mainly Labrador Retriever and Golden Retriever dogs are raised and trained.

A guide dog must be able to carry out various cognitive processes such as maintaining concentration, paying no attention to distractions, remembering its training, making decisions autonomously, and even ignoring its handler if they put their own life at risk [2,3]. Guide dogs are known to have a strong attachment or social connection to their handlers, as well as bilateral non-verbal communication [4,5]. This suggests that being a guide dog is demanding and exhausting and might require maintaining higher blood cortisol levels since increases in cortisol are positively associated with improvements in cognitive and emotional processes [6].

On the other hand, increases in cortisol are associated with the stress response in most mammals [6,7,8]. In dogs, cortisol can be measured in blood, saliva, hair, feces, and urine. While cortisol in hair and feces reflects its levels for several days to months, the evaluation of blood and saliva reflects acute increases experienced within the last few minutes [9]. In healthy dogs, salivary cortisol levels are the result of the passive diffusion of cortisol through the acinar cells of the salivary lumen [10,11]. Since the method for saliva collection is noninvasive, the measurement of salivary cortisol is a practical test in dogs for evaluating stress [10,11,12].

Since high cortisol levels are associated with the success of working dogs [13], it seems important and useful to know the stress levels that guide dogs maintain compared with dogs that have a similar breed but live as companion animals. Herein, we compared the salivary cortisol concentrations of active guide dogs with those of dogs that failed to achieve specialized guide dog training and are currently living as companion animals, which, to the best of our knowledge, is unknown information and may be useful for selecting guide dogs in the future.

## 2. Materials and Methods

### 2.1. Subjects

We included Labrador Retrievers and Golden Retrievers that were bred and trained at one GDTS in Mexico City. At three months of age, pups were adopted by foster families and raised over the next ten months. When dogs returned to the GDTS, they were evaluated to identify degenerative diseases and behavioral problems. When dogs showed dysplasia, aggression, fear, and anxiety, they were discarded from the guide dog training program (GDTP) and adopted by their foster families, becoming companion animals. The dogs that continued into the program (*n* = 14) received basic training. During this stage, the dogs lived at the GDTS facilities in individual kennels 3 m long × 1.6 m wide that had a covered area and an open area. Dogs were fed specialized Royal Canin^®^ (Gard, France) dry food twice daily with free access to water. After four months at the GDTP, 7 out of the 14 dogs successfully finished their training and began working as guide dogs, whereas the remaining dogs (*n* = 7) were discharged and adopted by their foster families to continue their lives as companion animals. The characteristics of the dogs included in the study are presented in Table 1.

Prior to the start of the study, approval from the Bioethics and Animal Welfare Commission of the Facultad de Medicina Veterinaria y Zootecnia, Universidad Veracruzana, was obtained (No.008/22). Before the start of the study, all authors completed an education program on the care and use of animals.

### 2.2. Experimental Procedure

We obtained and compared 4 saliva samples from all 14 dogs before, during, and after a period of social isolation and exposure to a 110-decibel sound capable of producing surprise and fear in animals [13]. On different days, the dogs, accompanied by their tutors (blind handlers or family members), were received at the GDTS between 9 and 12 h. Thirty minutes after arriving at the GDTS, the dog and its tutor entered a 3 m × 4 m room. Five minutes later, the first saliva sample was taken; then, the tutor was asked to leave the room, and the dog stayed, alone. Fifteen minutes later, a second saliva sample was taken, and immediately after that, a blank gun was fired to produce a 110-decibel sound. Fifteen and thirty minutes after the gun was fired, the third and fourth saliva samples were collected, respectively. Disposable cotton cords 10 cm long × 4 mm wide were used to obtain the saliva samples. The cord was inserted into the dog’s mouth while the handler held onto the other end. The animal was allowed to chew it for 60 s until moistened. The cord was cut into 3 cm pieces and inserted into a sterile 5 mL syringe, the plunger was pushed to extract the saliva, and it was collected in a 1.5 mL microtube. The samples were refrigerated (4 °C) for two hours and then frozen at −20 °C until processing [6]. To produce a 110-decibel sound, a Mendoza PK-62 sports gun was used. Salivary cortisol concentrations were assessed using a solid-phase immunoenzyme assay (ELISA) using the commercial kit Cortisol ELISA EIA-1887 (DRG^®^ International, Inc., Springfield, NJ, USA). The sensitivity of the assay was 2.5 ng/mL. The range of the curve was 2.5–200 ng/mL. The intra- and inter-assay coefficients of variation were 5.6% and 6.9%, respectively. Concentrations are expressed as ng/mL.

### 2.3. Statistical Analysis

A two-way ANOVA was used to identify significant differences between groups and between the number of saliva samples (1 to 4), as well as the interactions among them. The Fisher post hoc test was used to evaluate mean differences. The significance value for all comparisons was *p* < 0.05.

## 3. Results

There were significant differences between the guide dog and the companion dog groups: F (1, 3) = 16.31 and *p* < 0.001; the post hoc test showed that the guide dogs had higher levels of salivary cortisol than dogs living as companion animals (Figure 1). On the other hand, no differences were found between the sample times concerning the gunshot noise, F (1, 3) = 0.170 and *p* = 0. 916, or in the interaction between the group and saliva sample number, F (1, 3) = 0.97 and *p* = 0.979 (Figure 2).

## 4. Discussion

Basal cortisol levels are influenced by internal and external factors in an organism, such as age and sex and temperature and time of day, respectively [6,7]. However, stress, understood as a biological response of the organism caused by threats to its homeostasis, is the main cause of increased cortisol concentrations [6,7,8]. The hypothalamus–hypophysis–adrenal (HHA) axis is responsible for the release of cortisol as a consequence of stress [14,15]. In our study, guide dogs exhibited higher levels of cortisol (almost double) than dogs that live as companion animals. Research has shown that early training increases cortisol levels in dogs [16]. Furthermore, it has been observed that working dogs, such as those used in the military, increase cortisol levels when subjected to physical and sensory challenges [17]. While a companion dog lives without schedules, obligations, or social limits and with low demand for physical and mental activity, a guide dog is exposed daily to multiple and changing environmental stimuli and variable periods of activity and rest, requiring great concentration; sometimes, playing behavior is inhibited, and sometimes, it has to explore new territories, among other activities [1,4]. This suggests that guide dogs may have increased HHA axis activity as both a cause and a consequence of their daily activity. However, our results must be taken with caution since they represent only a small sample of animals and could be affected by factors such as genetics and environmental conditions in Mexico City, which is extremely noisy and crowded, and thus, they may not represent the reality for most guide dogs around the world.

Increases in cortisol levels are not necessarily bad but rather are necessary to better perform cognitive and emotional tasks such as those faced by a guide dog daily [18]. Studies have shown that individuals with hypoadrenocorticism, which produces low cortisol levels, show poor cognitive performance, poor sleep quality, low motivation, and mental fatigue [19]. On a typical day, a guide dog must maintain mental concentration for hours, ignore distractions, and make decisions, suggesting a high demand on its cognitive functions [3]. Based on this, we believe that the increase in cortisol in guide dogs compared with companion dogs is due to the high emotional and cognitive demands of adequately performing their work. Cortisol levels increase significantly in stressful periods compared with relaxation periods [20]. Social isolation is an important inducer of activity in the HHA axis and can increase cortisol release in animals, including humans [21,22]. In the dog, social isolation and separation from affection figures can lead to significant behavioral and physiological changes [23]. It is known that guide dogs form strong bonds with their handlers [24]; the guide dogs in our study had been interacting with their handlers for at least 1 year and did not separate from them at any time. Based on this assumption, it might be thought that greater levels of cortisol in guide dogs compared with companion dogs may be due to the separation period that the dogs had from their handlers during the test; however, the cortisol levels in both the guide and companion dogs were maintained throughout the test without increasing with the time the users were separated. Thus, we believe that social isolation and separation from their handlers had little to no effect on their cortisol levels. We must emphasize that, in our study, companion dogs do not represent the typical companion animal because, just like guide dogs, they belong to an elite group of dogs with suitable characteristics for work that received early stimulation from their very first days of life. Thus, we believe that a lack of cortisol increment in both groups of dogs, as a consequence of isolation or the gunshot, could be due to a remarkable resilience developed during their time at the GDTP.

Finally, neither the gunshot nor the isolation caused an observable effect on the cortisol levels of the dogs of either group. Given that all the dogs lived in Mexico City, where there is high noise pollution [25], we believe that the animals included in this study may be accustomed to noises of the same intensity as the gunshot and, therefore, showed no changes. Additionally, it is likely that the lack of response to noise in the dogs from both groups is because they received an auditory stimulation protocol when they were puppies when they belonged to the GDTP. We believe that increasing our understanding of guide dogs and working dogs may inspire further research into how the mind of this animal works and how we can improve selective breeding and training methods for the benefit of both dogs and humans.

## 5. Conclusions

In conclusion, the findings of this study suggest that the task of guiding individuals through various and unpredictable environments places significant demands on dogs. The results clearly demonstrate that guide dogs exhibited higher levels of cortisol compared with companion dogs. These results indicate that guide dogs consistently maintain elevated cortisol levels as a necessity or consequence of their working activity with blind individuals. The demanding nature of their role likely contributes to the heightened stress levels observed in these dogs. These findings emphasize the importance of recognizing and addressing the unique stressors faced by guide dogs to ensure their overall well-being and quality of life. Further research and investigation into effective stress management strategies for guide dogs are warranted to mitigate the potential negative impact of prolonged elevated cortisol levels. Ultimately, providing appropriate support and care for these remarkable animals is crucial to ensure their health, happiness, and ability to fulfill their important role as guides for the visually impaired.

## Figures and Tables

**Figure 1 animals-13-01981-f001:**
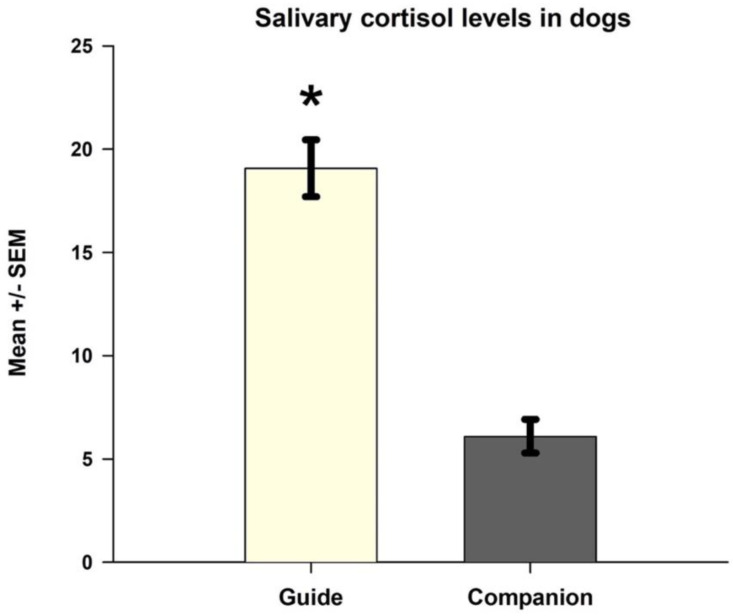
Guide dogs had higher salivary cortisol levels than companion dogs. * Indicates *p* < 0.05.

**Figure 2 animals-13-01981-f002:**
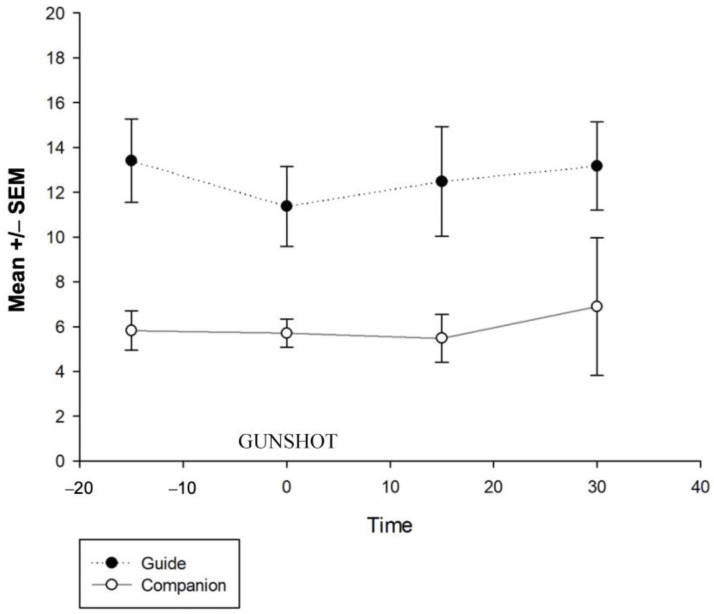
No differences were found between the sample times concerning the gunshot noise or in the interaction between the group and sample number. The first saliva sample was taken fifteen minutes before the gunshot (time 0). The second sample was collected 10 s after the gunshot, while the 3rd and 4th saliva samples were collected 15 and 30 min after the gun was fired.

**Table 1 animals-13-01981-t001:** Name, age, breed, and time living as guide or companion dogs. Years, y; months, m.

Guide Dogs				Companion Dogs			
Name	Age (Years)	Breed	Time as Guide Dog	Name	Age (Years)	Breed	Time as Companion Dog
Nusa	2	Golden retriever	1 y	Jock	4	Labrador retriever	2 y 4 m
Meli	5	Labrador retriever	3 y 2 m	Camila	6	Labrador retriever	4 y 3 m
Ninfa	2	Golden retriever	1 y	Gupy	7	Labrador retriever	4 y 9 m
Einy	7	Labrador retriever	6 y 2 m	Heidi	7	Labrador retriever	4 y 7 m
Ita	6	Labrador retriever	4 y 6 m	Elmo	7	Labrador retriever	5 y 7 m
Joe	4	Labrador retriever	1 y 7 m	Hunter	7	Labrador retriever	5 y 10 m
Fiona	7	Labrador retriever	5 y 8 m	Lancelot	7	Labrador retriever	3 y 2 m

## Data Availability

The authors prefer not to make the data available for public.

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
