# Peer review of "Salivary Cortisol in Guide Dogs"

_animals, 2023, doi:10.3390/ani13121981_

Round 1

Reviewer 1 Report

Summary

The paper evaluates salivary cortisol levels in guide and companion dogs during a period of social isolation and exposure to a gunshot sound. Results showed that 12 working guide dogs had significantly higher levels than the companion dogs. However, no changes were observed in relation to periods of social isolation or exposure to gunshot. Authors suggest the results support the conclusion that guide dogs maintain higher levels of basal cortisol due to their daily activity. Results will be of potential interest to guide dog organisations across the world.

General comments

The methods section could be enhanced through a further description of the dog demographics (e.g. sex) in each study group. As these dogs were all bred by one organisation levels of relatedness between the groups (e.g. littermate, unrelated) would also provide useful context for the reader.

The paper requires a limitations section within the discussion. This should include a discussion on the methods use, small sample size etc.

The paper would benefit from an application and future research section. This would include builds on current sample size and implications for assistance dog organisations.

Author Response

Dear reviewer 1
We have attended to all the observations and requests of the article.

A figure with description of the dogs have been added.

 Limitations and future  application section have been included.

Reviewer 2 Report

This study compares the salivary cortisol between guide dogs and previous guide dog candidates that were considered not suitable to become guide dogs. This is an interesting and important study considering the high demand and yet low success rate of guide dogs training. My suggestion about this manuscript is to clearly define the research question, aim and hypothesis. Accordingly, background information, research methodology and discussion should be tailored and focus more on the research question, aim and hypothesis. Also, there are many other factors found in previous literature that can influence guide dog cortisol. I would strongly encourage authors to investigate those studies before drawing the conclusion as there might be a risk of oversimplifying the research question and overinterpreting the study result.

Summary & abstract: “results showed that…It suggests that…” The results and conclusion in the summary and abstract need to be modified. What makes authors draw the conclusion based on the results? Something is missing here.

Summary & abstract: The “companion dogs” mentioned in the summary & abstract are not typical companion dogs, but dogs that were bred (and some trained) to become guide dogs though considered not suitable. This should be mentioned, or authors might consider using a different term to avoid readers’ confusion.

Line 40-41: “It suggests that working as a guide dog is demanding, exhausting, and requires a certain level of stress to succeed.” Please provide references. Also, please elaborate more about “requires a certain level of stress to succeed”. Not sure what does this mean.

Introduction: What is/are the aim(s) of this study? And corresponding hypotheses? The background information in the introduction should be around your aim(s) and hypotheses.

Experimental procedure: More background information is required to explain how the experimental procedure answers the question. In other words, what is the research question? And how the experimental procedure answers the research question? For instance, what is the correlation between the research question and exposing dogs under surprising/fear circumstances? This also goes back to my previous comment about the research aim and hypotheses. These should all be clearly defined and explained.

Discussion: There are many other publications about cortisol and guide dogs. Some of them have contradicted results or proposed that other external factors might have greater effects on guide dog cortisol. Please give a more thorough discussion around this as this seems to be the main purpose of this paper. Be careful not to oversimplify the research question and overinterpret the study result.

The language is fine and I have no issue reading the manuscript. 

Author Response

Summary & abstract: “results showed that…It suggests that…” The results and conclusion in the summary and abstract need to be modified. What makes authors draw the conclusion based on the results? Something is missing here. Information has been added and clarify in the abstract.

Summary & abstract: The “companion dogs” mentioned in the summary & abstract are not typical companion dogs, but dogs that were bred (and some trained) to become guide dogs though considered not suitable. This should be mentioned, or authors might consider using a different term to avoid readers’ confusion. Comments related to the characteristics of companion dogs used in the study have been added.

Line 40-41: “It suggests that working as a guide dog is demanding, exhausting, and requires a certain level of stress to succeed.” Please provide references. Also, please elaborate more about “requires a certain level of stress to succeed”. Not sure what does this mean.Cortisol hormones, produced by stress, improve memory. Animals can stay focused, and memory is sharper. Cortisol allows above average productivity. That is why we propose that guide dogs require a certain level of stress to be successful in their daily activities.

Introduction: What is/are the aim(s) of this study? And corresponding hypotheses? The background information in the introduction should be around your aim(s) and hypotheses. Objective and research question has been included.

Experimental procedure: More background information is required to explain how the experimental procedure answers the question. In other words, what is the research question? And how the experimental procedure answers the research question? For instance, what is the correlation between the research question and exposing dogs under surprising/fear circumstances? This also goes back to my previous comment about the research aim and hypotheses. These should all be clearly defined and explained.

Discussion: There are many other publications about cortisol and guide dogs. Some of them have contradicted results or proposed that other external factors might have greater effects on guide dog cortisol. Please give a more thorough discussion around this as this seems to be the main purpose of this paper. Be careful not to oversimplify the research question and overinterpret the study result.

To our knowledge there are no published studies on salivary cortisol levels in active guide dogs. There are many other publications about cortisol in working dogs, but they do not include guide dogs. The only one study related to salivary and guide dogs, explored the effect of human interaction and isolation on oxytocin and cortisol levels in eight guide dogs. They found that oxytocin concentrations showed a statistically significant increase after the positive interaction, but no difference was found between the cortisol concentrations after each experimental condition.

The discussion has been increased and we consider that the doubts have been clarified.

We believe that our study is novel, original, and will be of interest to the readers of the journal.

Round 2

Reviewer 2 Report

1. It is still not clear to me why the findings lead to the conclusion, "...guide dogs maintain higher cortisol levels as requirement or consequence of their working activity with blind people." What do you mean "requirement"? Does that mean guide dogs need a higher cortisol concentration for their tasks? Or perhaps guide dogs are just being aroused most of the time due to the job nature? How do you know the cause and effect?

2. Why did you use such an experimental procedure that included social isolation and exposure to sound? Were they just to simulate a stressful situation or to simulate some real-life scenario? In other words, how did the experimental design answer your research questions?

Minor editing of English language required.

Author Response

  1. It is still not clear to me why the findings lead to the conclusion, "...guide dogs maintain higher cortisol levels as requirement or consequence of their working activity with blind people." What do you mean "requirement"? Does that mean guide dogs need a higher cortisol concentration for their tasks? Or perhaps guide dogs are just being aroused most of the time due to the job nature? How do you know the cause and effect?

We have modified the conclusion, so the word “requirement” has been removed. It has been modified in introduction, discussion, and conclusion.

In a typical day, a guide dog must stay focused or concentrated for hours, ignoring distractions and making decisions, which suggests high cognitive demand. 

We believe that the increase in cortisol in guide dogs compared to companion dogs is due to the high emotional and cognitive demand they need to adequately perform their job.

  1. Why did you use such an experimental procedure that included social isolation and exposure to sound? Were they just to simulate a stressful situation or to simulate some real-life scenario? In other words, how did the experimental design answer your research questions?

Social isolation and exposure to sound are within the most common and valid experimental procedures to assess temperament and emotion in dogs. In our study we decided to use them to identify differences in the cortisol response between guide dogs and companion dogs.

Round 3

Reviewer 2 Report

I agree with the summary, "...It suggests that guide dogs maintain higher levels of basal cortisol compared to companion dogs, which could be associated to cognitive processes derived from working as guide dogs." However, in the conclusion, authors stated "...It indicates that higher cortisol levels can be a good indicator of guide dog success and can be used as criteria for 247 selective breeding."

I am not sure about the conclusion regarding high cortisol levels as an indicator of guide dog training success. There is a correlation between high cortisol concentrations and being a guide dog, but this does not justify it as an indicator of training success. Please do not overinterpret the results. 

Author Response

Dear Reviewer

The conclusions have been restated as follows:

In conclusion, the findings of this study suggest that the task of guiding individuals through various and unpredictable environments places significant demands on dogs. The results clearly demonstrate that guide dogs exhibited higher levels of cortisol compared to companion dogs. These results indicate that guide dogs consistently maintain elevated cortisol levels as a necessity or consequence of their working activity with blind individuals. The demanding nature of their role likely contributes to the heightened stress levels observed in these dogs. These findings emphasize the importance of recognizing and addressing the unique stressors faced by guide dogs to ensure their overall well-being and quality of life. Further research and investigation into effective stress management strategies for guide dogs are warranted to mitigate the potential negative impact of prolonged elevated cortisol levels. Ultimately, providing appropriate support and care for these remarkable animals is crucial to ensure their health, happiness, and ability to fulfill their important role as guides for the visually impaired.